# ATTRIBOT: A BAG OF TRICKS FOR EFFICIENTLY APPROXIMATING LEAVE-ONE-OUT CONTEXT ATTRIBUTION

**Fengyuan Liu** [*] **Nikhil Kandpal, Colin Raffel**
University of Toronto, Vector Institute
Toronto, Ontario, Canada
`fy.liu@mail.utoronto.ca`

## ABSTRACT

The influence of contextual input on the behavior of large language models (LLMs) has prompted the development of *context attribution* methods that aim to quantify each context span's effect on an LLM's generations. The leave-one-out (LOO) error, which measures the change in the likelihood of the LLM's response when a given span of the context is removed, provides a principled way to perform context attribution, but can be prohibitively expensive to compute for large models. In this work, we introduce AttriBoT, a series of novel techniques for efficiently computing an approximation of the LOO error for context attribution. Specifically, AttriBoT uses cached activations to avoid redundant operations, performs hierarchical attribution to reduce computation, and emulates the behavior of large target models with smaller proxy models. Taken together, AttriBoT can provide a $>300\times$ speedup while remaining more faithful to a target model's LOO error than prior context attribution methods. This stark increase in performance makes computing context attributions for a given response $30\times$ faster than generating the response itself, empowering real-world applications that require computing attributions at scale. We release a user-friendly and efficient implementation of AttriBoT to enable efficient LLM interpretability as well as encourage future development of efficient context attribution methods [1].

## 1 INTRODUCTION

The use of large language models (LLMs) has proliferated in recent years including the integration of OpenAI's GPT-4 (OpenAI, 2024) and Google's Gemini (Team, 2024) into Apple and Android-based products with billions of users. As LLMs become more widely used, their influence on information access, decision-making, and social interactions will grow, as will with the consequences of incorrect or problematic outputs. The risks and impact of this widespread adoption spur the need for a deeper understanding of *how* and *why* LLMs generate their outputs. Indeed, a great deal of recent work on LLM interpretability aims to uncover and elucidate their inner workings, including determining the influence of pre-training data (Koh & Liang, 2020; Grosse et al., 2023; Kandpal et al., 2023) and mechanistically understanding their underlying architecture (Cammarata et al., 2020).

A common usage pattern for LLMs involves providing relevant contextual information alongside a query. For example, in retrieval-augmented generation (RAG) (Lewis et al., 2021), documents from an external datastore that are relevant to a given query are retrieved and are provided as part of the LLM's input. This allows LLMs to process data or make decisions based on information that is not available in their pre-training dataset, which has proven critical to making LLMs applicable to a wide range of use cases. While inspecting the documents retrieved by a RAG system can provide a form of interpretability, LLMs generally provide no direct insight into which part of the augmented context influenced the model's generation. To address this shortcoming, *context attribution* methods

---

[*] Work done at Vector Institute
[1] https://github.com/r-three/AttriBoT

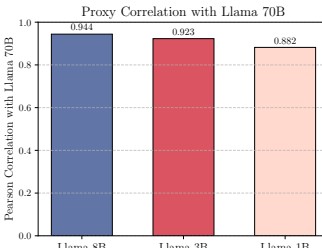 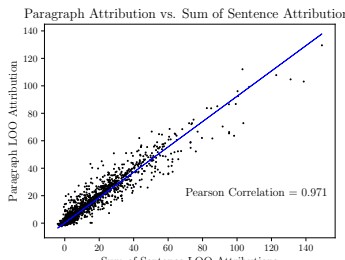 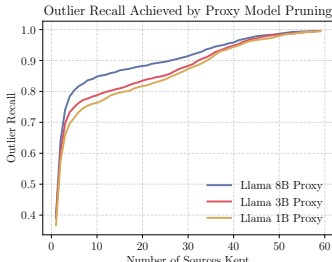

Figure 1: We empirically test the assumptions underlying the AttriBoT's underlying methods on examples from Hotpot QA. **Left:** The attribution scores of small proxy models ranging from 1B to 8B parameters have high correlation with the attribution scores of a 70B-parameter target model, implying that the attributions from smaller models can be a reliable proxy for those from a target model. **Middle:** Paragraph-level attribution scores correlate extremely well ($R = 0.97$) with the sum of the sentence-level attribution scores in a given paragraph, suggesting that hierarchical attribution can provide an effective means of pruning a large amount of irrelevant context. **Right:** Proxy models can effectively prune contexts of unnecessary sources, achieving recall of $90\%$ when keeping only half of the sources in a context.

(Cohen-Wang et al., 2024; Yin & Neubig, 2022; Gao et al., 2023b) aim to quantify the influence of each span of text in an LLM's context on its generated output.

A natural approach for context attribution is to remove a span of text from the context and measure the ensuing change in the likelihood of the model's original response. This notion of importance, known as the Leave-One-Out (LOO) error, is a common idea used in training data attribution (Koh & Liang, 2020), data valuation (Choe et al., 2024), feature attribution (Li et al., 2017), and recently for context attribution (Cohen-Wang et al., 2024). While the LOO error produces meaningful and interpretable context attribution scores, it is often viewed as impractical due to the need to perform an independent forward pass to score each text span in the context. This is particularly problematic for modern LLMs whose forward passes are computationally expensive and realistic use cases often involve long contexts with many text spans (e.g., sentences or paragraphs) to score.

In this work, we aim to show that LOO context attributions can be efficiently approximated at LLM scale. To do so, we leverage the following observations:

1. Approximately half of the FLOPs needed to naïvely compute LOO attributions are redundant and can be avoided by caching the attention key and value tensors at each layer (Pope et al., 2022).

2. The sum of the LOO attributions for $k$ contiguous text spans (e.g., sentences in a paragraph or paragraphs in a section) are well-approximated by a single Leave-$k$-Out attribution score (Figure 1, middle).

3. The LOO attribution scores for a large model (e.g., 70B parameters) are well-approximated by smaller models (e.g., 8B parameters) in the same model family (Figure 1, left and right).

These findings naturally lend themselves to efficient key-value caching schemes, novel hierarchical approaches that prune low-information context chunks before performing attribution, and methods that leverage small proxy LMs to approximate the LOO attributions of a larger target model. We develop and explore the practical application of these schemes to develop an aggregate system called AttriBoT (a **B**ag **o**f **T**ricks for efficient context attribution). When evaluating AttriBoT in the open-book question answering setting, where a model is presented a question with one or more related documents provided in its context, we find that our methods significantly reduce the cost of computing LOO attributions – at times by $>300\times$ – while remaining more faithful to the original model's LOO error than a wide range of baselines. In addition, the cascade of approaches underlying AttriBoT can be naturally composed to attain a Pareto-optimal trade-off between efficiency and accuracy over efficiencies ranging multiple orders of magnitude.

In the following section, we formalize the problem of context attribution, define the LOO error in detail, and discuss the metrics we use to evaluate a given approximate attribution method's faithfulness. Then, in Section 3, we detail our novel techniques for efficiently approximating LOO error,

ultimately producing AttriBoT. Experimental results across multiple model families and datasets are provided in Section 4.1, with related work in Section 5 and a conclusion in section 6.

## 2 PROBLEM STATEMENT

In this section, we introduce the problem of context attribution (Section 2.1), define the simple and principled Leave-One-Out (LOO) error method for context attribution (Section 2.2), and define how we evaluate approximate LOO attribution methods (Section 2.3).

**Notation** Suppose that an autoregressive Transformer (Vaswani et al., 2023) language model (e.g. Llama (Dubey et al., 2024), GPT-4 (OpenAI, 2024), etc.) generates a response $R$ conditioned on a query $Q$ and a context $C$. For the purposes of this paper, we view a language model with parameters $\theta$ as a function $p_\theta(R|Q, C)$ that returns the probability of generating a response given a particular query and context. Furthermore, we assume the context can be partitioned into a sequence of sources with a partitioning function $\Pi(C) = [s_i]_{i=1}^{N}$, where each source $s_i$ is a contiguous chunk of text in the context, like a sentence, paragraph, or document. For notational convenience, we define $|C|$ to be the number of sources in $\Pi(C)$.

### 2.1 CONTEXT ATTRIBUTION

A large body of work on feature attribution has studied the relationship between a model's predictions and input features (Li et al., 2016; Wu et al., 2021). More recently, the problem of *context attribution*, coined by Cohen-Wang et al. (2024), has been introduced as a special case of feature attribution, where a response generated by an LLM is attributed back to parts of the LLM's contextual information.

Formally, we follow Cohen-Wang et al. (2024) and define a context attribution method as a function $\tau(\theta, R, s_1, ..., s_{|C|}) \in \mathbb{R}^{|C|}$ that maps a language model's parameters $\theta$, response $R$, and sources $s_i \in \Pi(C)$ to a vector of real-valued scores indicating each source's importance to the model's response. While Cohen-Wang et al. (2024) leave the dependence of $\tau$ on $\theta$ and $R$ implicit, we make this dependence explicit because we will later explore algorithms where multiple models are used and may be fed other models' responses.

### 2.2 LEAVE-ONE-OUT ATTRIBUTION

There are many notions of what makes a source important that each lead to different choices of $\tau$. Perhaps the simplest and most interpretable notion of importance is that important sources lead to large changes in the likelihood of the model's response when they are removed from the original context. This quantity, often referred to as the Leave-One-Out (LOO) error, leads to the following choice of context attribution function, which we refer to as the "LOO attribution" in the remainder of the paper:

$$\tau_{LOO}(\theta, R, s_1, ..., s_{|C|})_i = \log p_\theta(R|Q, C) - \log p_\theta(R|Q, C \setminus \{s_i\}) \qquad (1)$$

In practice, LOO attributions can be impractical to compute for particularly large language models since scoring all $|C|$ sources in the context requires $|C| + 1$ forward passes of the language model – one pass to compute the likelihood of the response given the full context and $|C|$ passes to compute the likelihood with each of the $|C|$ sources individually removed from the context. In realistic settings where models are provided with a large amount of contextual information (i.e. $|C|$ is very large), computing LOO attributions for all sources in a context can be orders of magnitude more expensive than generating the response itself. Thus, the remainder of this paper explores a variety of methods for efficiently approximating $\tau_{LOO}$.

### 2.3 EVALUATING APPROXIMATE LEAVE-ONE-OUT ATTRIBUTIONS

Much past work on intrepretability and attribution measures performance in terms of how well a given method matches human annotations of source importance. Our focus is instead on efficiently approximating the attributions of a "target" LLM whose LOO attributions are prohibitively expensive to compute. We therefore introduce a straightforward evaluation procedure for measuring effi-

| Method | Method Parameters | Theoretical FLOPs | Speedup over LOO |
|---|---|---|---|
| LOO | N/A | $2PT\|C\|(\|C\|-1)$ | 1 |
| KV Caching | N/A | $PT\|C\|(\|C\|-1)$ | 2 |
| Proxy | $P'$ : Proxy model size | $2P'T\|C\|(\|C\|-1)$ | $\frac{P}{P'}$ |
| Pruning | $P'$ : Proxy model size $\alpha$ : Fraction of sources | $2PT\|C\|((\alpha^2 + \frac{P'}{P})\|C\| - \alpha - \frac{P'}{P})$ | $\frac{P(\|C\|-1)}{(\alpha^2 P + P')\|C\| - \alpha P - P'}$ |
| Hierarchical | $H$ : Sources per group $\beta$ : Fraction of groups | $2PT\|C\|((\beta^2 + \frac{1}{H})\|C\| - \beta - 1)$ | $\frac{H(\|C\|-1)}{(\beta^2 H+1)\|C\| - \beta H - H}$ |

Table 1: The theoretical number of floating-point operations (FLOPs) needed by different methods to compute LOO attributions expressed in terms of $P$, the number of target model parameters, $T$, the number of tokens per source, and $C$, the number of context sources.

ciency and determining how faithful an approximate method's attributions are to those produced by the original target model.

**Approximation Error** Past work has found that for many real-world tasks like summarization and question answering, attributions tend to be sparse, meaning that only a small number of sources in the context significantly influence a model's response (Liu et al., 2024; Kuratov et al., 2024; Yang et al., 2018). Thus, when evaluating how well an algorithm approximates LOO attributions, we are primarily interested in how well the algorithm recovers these few highly contributive sources.

To evaluate approximate LOO attribution methods we first compute the LOO attributions of the target model and identify the most contributive sources. We consider a source to be highly contributive if it has an outlyingly large score. To find outlier scores, we apply the Generalized Extreme Studentized Deviate (ESD) test (Rosner, 1983), a statistical test that returns the outlier values in a collection of data (for more details, see Appendix B.2). We denote the number of detected outlier sources for a given example as $n_{out}$.

We then test the extent to which approximate LOO attribution methods recover these $n_{out}$ outlier sources using evaluation metrics from information retrieval. In particular, we rank the sources according to their approximate LOO attribution scores and measure the mean Average Precision (mAP) over examples in a dataset (Manning et al., 2008).

**Efficiency** For each approximate LOO algorithm, we report its practical and theoretical efficiency. To measure practical efficiency, we report the average GPU-seconds needed to compute attribution scores for all sources in a context. For theoretical efficiency, we estimate the number of floating-point operations (FLOPs) needed to compute attributions as a function of the number of model parameters $P$, the number of sources in the context $|C|$, the number of tokens in each source $T$, and any other method-specific parameters. To estimate FLOPs, we use the simplifying assumptions from Kaplan et al. (2020) and Hoffmann et al. (2022), i.e. that performing a forward pass for a $P$-parameter language model on $T$ tokens requires approximately $2PT$ FLOPs.

## 3 Accelerating Leave-One-Out Attribution with AttriBoT

In this section, we describe the **B**ag **o**f **T**ricks that AttriBoT uses to efficiently compute approximate LOO attributions. For each method, we provide intuition for why it serves as an effective approximation to the LOO error as well as discussion on its theoretical speedup over standard LOO attribution using the target model. Full details on the efficiency gains for each method can be found in Table 1 with derivations available in Appendix A.

### 3.1 Key-Value Caching

For a context containing $|C|$ sources, computing $\tau_{LOO}$ requires $|C| + 1$ forward passes (including the first forward pass to generate the response). However, much of this computation is redundant when the underlying language model is an autoregressive Transformer. At each self-attention layer

in an autoregressive Transformer, the key and value tensors for a given position in the sequence are only a function of previous tokens due to autoregressivity or causal masking (Vaswani et al., 2023). Thus, if two inputs share a prefix, the key and value tensors at each position in the prefix will be identical. This redundant computation can be avoided by caching the key and value tensors at each layer and reusing the cached values for inputs that have a previously computed prefix (Pope et al., 2022).

Specifically, when computing LOO attributions, the keys and values can be cached for the full sequence of tokens while computing $p_\theta(R|Q, C)$. Then, for each subsequent forward pass computing $p_\theta(R|Q, C \setminus \{s_i\})$, the cached keys and values for $Q$ and the first $i-1$ sources in $C$ can be reused. In aggregate over all sources, this avoids computation for a total of $(|C|-1)/2$ sources, ultimately saving approximately half of the FLOPs used to compute $\tau_{LOO}$. Notably, ignoring differences due to numerical error of floating point operations, we should expect KV caching to be lossless – i.e., LOO attributions should be the same regardless of whether KV caching is used.

## 3.2 HIERARCHICAL ATTRIBUTION

In many settings, we expect that the context has a hierarchical nature – for example, the context might comprise a sequence of paragraphs that can each be broken down into a sequence of sentences. This structure can be leveraged to efficiently identify sources that are likely to have high LOO attributions, allowing us to avoid computing $\tau_{LOO}$ for every source in the context.

Imagine we want to compute LOO attributions at the sentence level. The key assumption underlying our hierarchical attribution algorithm is that the sum of the LOO attributions for $k$ sentences in a paragraph can be closely approximated by a single Leave-$k$-out attribution score computed by removing the paragraph as a whole. In cases where this holds, paragraphs whose removal incurs a large drop in the response's likelihood are also likely to contain highly contributive sentences.

Specifically, to perform hierarchical attribution we assume the context can be partitioned into a sequence of "source groups" (e.g., paragraphs) $\Pi_g(C) = [G_i]_{i=1}^M$ and each of these groups can be further decomposed into its constituent sources (e.g., sentences) $\Pi_s(G_i) = [s_j]_{j=1}^{M_i}$, where $M_i$ is the number of sentences in group $G_i$. We first compute the LOO error for each source group, $\tau_{LOO}(\theta, R, G_i, \ldots, G_M)$, and keep only a fraction, $\beta$, of the groups with the highest scores. From these high-attribution groups, we construct a shortened context $\hat{C}$ comprising only the retained groups and compute $\tau_{LOO}(\theta, R, s_1, \ldots, s_{|\hat{C}|})$ for the sources $s_i \in \Pi_s(\hat{C})$.

For example, when processing a document with 10 paragraphs and using $\beta = 0.2$, we would first compute the LOO error corresponding to the removal each of the paragraphs and retain the two paragraphs with the highest LOO score. Finally, to compute attributions for the sentences in these two paragraphs, we concatenate the paragraphs into a truncated context and compute the LOO error incurred by removing each sentence from the truncated context.

This hierarchical approach achieves a speedup over LOO by (1) first doing fewer forward passes over the full context when computing attributions at the source group level and (2) doing fewer forward passes on a shortened context after the low-attribution source groups are removed. With the simplifying assumption that each source group contains $H$ sources, if we keep only a constant number source groups irrespective of the total number of sources (i.e. $\beta \sim 1/|C|$), then this method's speedup over LOO is approximately $H$ for long contexts.

## 3.3 PROXY MODELING

The high cost of computing the LOO error stems from having to compute $|C|$ forward passes through a large and computationally expensive language model. The cost of each of these forward passes is naturally cheaper for smaller models. If a smaller "proxy" model's attributions are faithful to a larger target model whose attributions we hope to obtain, this raises the possibility of reducing the cost of computing LOO attributions by using the proxy model instead. In particular, we might hope that a smaller model from the same model family (i.e. sharing a model architecture, training dataset, and training objective but differing in its parameter count) produces similar attributions to a target model. Specifically, we take the response $R$ generated by the target model, and use the approximation $\tau_{LOO}(\theta_{proxy}, R, s_1, \ldots, s_{|C|}) \approx \tau_{LOO}(\theta, R, s_1, \ldots, s_{|C|})$. The proxy model does

not need to produce the response, but rather emulate the conditional likelihood of the target model's response given a context. In practice, as further discussed in section 4.2.1, we find that when using the target model's response to compute attribution scores, the LOO errors from a target model are highly correlated with those from a small proxy model from the target model.

## 3.4 PROXY MODEL PRUNING

While small proxy models can approximate the context attributions of larger target models reasonably well, one way to improve the fidelity of this approximation is to use the proxy model to prune away low-attribution sources and then re-score the remaining sources with the target model. Specifically, we take the response generated by the target model, use it to compute $\tau_{LOO}(\theta_{proxy}, R, s_1, \ldots, s_{|C|})$, and keep only a fraction, $\alpha$, of the highest scoring sources. Then we reconstitute the context keeping only these top sources and recompute their LOO attributions using the target model.

Compared to simply using a proxy model, this method is more expensive since it requires forward passes of the target model for each of the non-pruned spans. However, this extra compute is spent improving the approximation error for the high-attribution sources whose LOO attributions are most important to recover accurately. If the number of sources kept after pruning is constant (i.e. $\alpha \sim 1/|C|$), then this method provides a speedup of roughly $P/P'$ for long contexts, the same speedup achieved by proxy modeling alone.

## 3.5 COMPOSING METHODS IN THE ATTRIBOT BAG OF TRICKS

The methods described above improve the efficiency of computing LOO attributions by avoiding redundant calculations during forward passes, reducing the size of the model used to perform the forward passes, and/or reducing the total number of forward passes. As different methods target different factors that make exact LOO computation expensive, these methods can often be composed together to achieve even greater efficiency. Concretely, in addition to the individual methods described previously, we test the following composite methods in our experiments: KV Caching + Hierarchical Attribution, KV Caching + Proxy Modeling, KV Caching + Proxy Model Pruning, and KV Caching + Proxy Modeling + Hierarchical Attribution.

## 4 EXPERIMENTS

To validate AttriBoT, we compare its efficiency and faithfulness to a range of baseline context attribution methods. Our focus is on confirming that our approximations provide a Pareto-optimal trade-off between speed and faithfulness to the target model.

## 4.1 SETUP

In this section, we describe the models, datasets, and baseline methods used to evaluate AttriBoT, as well as the computational resources used to run our experiments.

### 4.1.1 MODELS

To ensure that AttriBoT is valuable in state-of-the-art settings, we focus on recent performant open LLMs. We therefore experiment with two target models whose LOO context attributions we aim to efficiently approximate: the 70B-parameter instruction-tuned LLM from the Llama 3.1 model family (Dubey et al., 2024) and the 72B-parameter instruction-tuned LLM from the Qwen 2.5 model family (Yang et al., 2024). As proxy models, we use smaller instruction-tuned models from each of these model families. For Llama 70B, this includes the 8B-parameter model from the Llama 3.1 family and the 1B and 3B-parameter models from the Llama 3.2 family. For Qwen 72B, we use the 0.5B, 1.5B, 3B, 7B, and 32B-parameter models from the Qwen 2.5 model family as proxies. Each model within a given family shares an architecture, pre-training objective, and pre-training dataset. We report results for the Llama models in the main text and replicate select experiments with the Qwen models in Appendix C.

### 4.1.2 DATASETS

There are a number of applications of context attribution that range from verifying faithfulness of model summaries to detecting malicious prompts. In this study, we focus on the well-studied and widespread setting of open-book question answering (OBQA), which involves prompting a model to answer a question that is supported by a provided context. Since many tasks can be formulated as answering a question about contextual information (McCann et al., 2018), we expect results on OBQA to be a reliable indicator of performance in other settings. Given that each method's efficiency depends on the context length, we consider datasets covering a range of context lengths. To this end, we focus on three open-book QA datasets:

1. SQuAD 2.0 (Rajpurkar et al., 2018): A reading comprehension benchmark where a model is presented a question along with a Wikipedia article containing the answer. For this dataset, we consider sources to be individual sentences in the context and source groups for hierarchical attribution to be paragraphs.

2. HotpotQA (Yang et al., 2018): A multi-hop question answering benchmark that requires reasoning across paragraphs from multiple Wikipedia pages, as well as ignoring distractor paragraphs, in order to answer a question. For this dataset, we consider sources to be individual sentences in the context and source groups for hierarchical methods to be paragraphs.

3. QASPER (Dasigi et al., 2021): A document-grounded, information-seeking question answering dataset where each context is a natural language processing paper. Answering questions in the QASPER dataset requires complex reasoning about claims made in multiple parts of a paper. For this dataset, we consider sources to be paragraphs and source groups for hierarchical methods to be sections of the paper provided in the context.

For more details on these datasets, see Appendix B.1.

### 4.1.3 BASELINES

We compare AttriBoT's performance to a diverse set of context attribution methods:

1. Attention Weights: The self-attention operation in the Transformer architecture provides an implicit way of determining which input entries are influential (Jain & Wallace, 2019; Wiegreffe & Pinter, 2019). Concretely, we compute the total attention weight for each source by summing the attention weights of a source's tokens across all attention heads and layers.

2. Gradient Norm: The gradient of the model's response likelihood with respect to its input provides a first-order approximation of the model's sensitivity input perturbations, and thus is a useful interpretation technique (Yin & Neubig, 2022; Mohebbi et al., 2021). Specifically, we calculate the Frobenius norm of the gradient of the response's likelihood with respect to the token embeddings of each source.

3. Sentence embedding similarity: As a simple model-agnostic baseline, we compute the similarity between sentence embeddings for the generated response and each one of the sources. We generate sentence embeddings using the all-MiniLM-L6-v2 SentenceBERT model (Reimers & Gurevych, 2019) and compute similarity via cosine similarity.

4. ContextCite: This method estimates the importance of each context source by training a linear surrogate model to estimate the likelihood of a response given a set of context sources. The trained linear model's weights for each source in the context act as an attribution score (Cohen-Wang et al., 2024). More background on ContextCite is provided in in Appendix B.3.

For more details on baseline methods, see Appendix B.3.

### 4.1.4 COMPUTATIONAL RESOURCES

All experiments were run on servers with 4 NVIDIA A100 SXM4 GPUs with 80GB of VRAM. For experiments involving models with fewer than 15 billion parameters, we use only a single GPU. For models with more parameters, we use model parallelism across multiple GPUs.

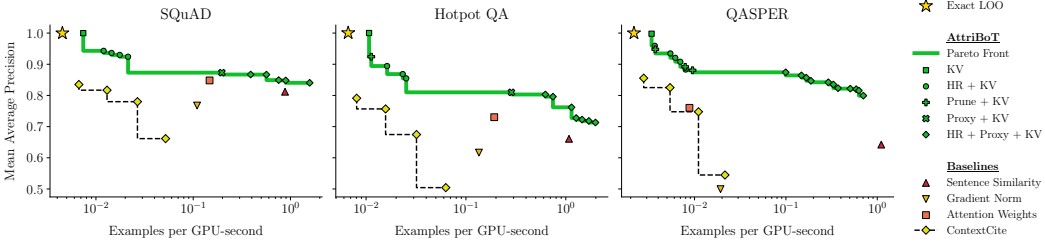

Figure 2: We plot the mean average precision compared to attributions of the target model against the GPU time for AttriBoT and a variety of baselines using Llama 3.1 70B Instruct as the target model and smaller Llama instruct variants as proxy models. Across all three datasets, AttriBoT is consistently Pareto-optimal over multiple orders of magnitude.

## 4.2 RESULTS

We now provide empirical results that motivate the approximations made in AttriBoT and demonstrate its effectiveness in the OBQA setting.

### 4.2.1 VALIDATING ATTRIBOT'S APPROXIMATIONS

**Hierarchical Attribution**   The assumption underlying our hierarchical attribution method is that the sum of the LOO scores for $k$ sources in a source group (e.g., sentences in a paragraph) can be estimated by computing the Leave-$k$-Out attribution of the source group as a whole. We empirically test this assumption for Llama 70B by comparing LOO attributions for all paragraphs in HotpotQA with the sum each paragraph's sentence-level LOO attributions. Figure 1 (Middle) shows that the these two quantities have a very high Pearson correlation ($R = 0.97$), meaning that the removal of low-attribution source groups in our proposed hierarchical attribution method is likely to keep most high-attribution sources.

**Proxy Modeling**   Proxy modeling assumes that two models from the same model family will have similar LOO attributions. Thus, if the target model is large, its LOO attributions for a particular response can be efficiently approximated by attributing the same response with a smaller proxy model. We test this claim by measuring the correlation between the HotpotQA sentence-level attribution scores from Llama 70B and smaller proxy models from the Llama model family, ranging from 1B to 8B parameters. We find that LOO attributions from each of the proxy models are very well correlated with the Llama 70B's LOO attributions. Shown in Figure 1 (Left) the Pearson correlation is as high as 0.94 for the largest proxy model with 8B parameters and only decreases to 0.88 for the smallest 1B-parameter proxy model.

**Proxy Model Pruning**   Like proxy modeling, proxy model pruning assumes that small models can capably approximating a large model's attributions. However, pruning additionally requires the proxy model to rank outlier sources as one of the top-$\alpha|C|$ sources. In practice we find this to be the case. Figure 1 (Right) plots the outlier recall of different proxy models as a function of the number of sources kept when pruning a context. In practice, we find that Llama 8B achieves $85\%$ recall of sources with outlier Llama 70B attributions while only keeping the top-10 spans.

### 4.2.2 ATTRIBOT'S PARETO-OPTIMALITY

Our main results, comparing AttriBoT to the baselines listed in Section 4.1.3 on the datasets described in Section 4.1.2 for the Llama model family, are provided in Figure 2. We find that the techniques composed in AttriBoT produce a Pareto-optimal method across many orders of magnitude of costs on all three of our evaluation datasets. Compared to ContextCite, the only other method we are aware of that computes attributions via repeated forward passes on modified contexts, AttriBoT can provide attributions that are equally faithful to a target model's LOO attributions with orders of magnitude lower cost. Compared to cheaper attribution baselines (sentence similarity, gradient norm, and attention weights), AttriBoT remains Pareto-optimal and can even be more efficient than

using sentence embeddings (which are produced by a small model and do not require any forward passes of a target or proxy model). These results hold true across all datasets, demonstrating that AttriBoT scales well to datasets of varying context length.

**KV Caching** We find that, in practice, KV caching offers about a $1.6\times$ speedup over computing exact LOO attributions while, as expected, nearly perfectly recovering outlier sources (see Figure 3. While in theory computing attributions with and without KV caching should be identical, in practice we observe a small difference in the attribution values due to accumulated numerical error of floating-point operations.

**Proxy Modeling** Shown in Figure 3, proxy modeling can be an effective approach for greatly speeding up LOO computation. We find that as the size of the proxy model decreases, its faithfulness to the exact LOO attributions also decreases. However, the faithfulness of attributions from smaller models still remains better than many baselines that use orders of magnitude more compute.

**Hierarchical Attribution** In Figure 4, we show the accuracy vs. efficiency trade-off of hierarchical attribution. We find that modulating the fraction of source groups retained, $\beta$, is an effective method for trading off accuracy for efficiency. Additionally, we find that using the target model yields the most accurate attributions, while substituting the target model for a smaller proxy model achieves greater efficiency.

**Proxy Model Pruning** Like hierarchical attribution, we show in Figure 5 that modulating the fraction of sources to retain, $\alpha$, and the proxy model size effectively trades off accuracy for efficiency.

### 4.2.3 ATTRIBOT'S IMPACT ON MATCHING HUMAN ANNOTATIONS

Our main focus has been on efficiently approximating the attributions of a target model. However, it can also be valuable to measure how closely an attribution method matches human-annotated "important" spans. Such annotations are available for the HotpotQA dataset, so we additionally evaluated the impact of using AttriBoT on agreement with human annotations. Full results are in Appendix C.4; as a short summary, we find that the methods introduced in AttriBoT actually remain *more* faithful to the human-annotated attributions than they do to those of the target model. Concretely, we observe that AttriBoT can achieve a $300\times$ speed up with only 10% drop in mAP.

## 5 RELATED WORK

**Generating Citations** Computing LOO context attributions involves repeated inference of the target model. In contrast, past work has explored whether it is possible to train LLMs to directly provide citations to sources or spans from their context (Weller et al., 2024; Gao et al., 2023b). Some approaches leverage pipelines for retrieving external facts to ground and verify a generated response while providing explicit attribution (Asai et al., 2023; Li et al., 2024b; Sun et al., 2023; Li et al., 2024a; Chen et al., 2024; Huo et al., 2023; Gao et al., 2023a). Other models are explicitly trained to produce citations, including GopherCite (Menick et al., 2022), which generates quotes from supporting documents after generating a response; LaMDA (Thoppilan et al., 2022), which provides URLs from the information retrieval system used; and WebGPT (Nakano et al., 2022), which provides extracted context from the source webpages. Such citation generation methods only provide insight how each statement is possibly supported by the corresponding source instead of measuring how each source contributes to a response via a causal intervention like source removal. Therefore, while useful for providing evidence for supporting or verifying an LLM's response, these methods cannot necessarily provide insight into *why* an LLM produced a particular output.

While one goal of this line of work is to improve the factuality and reduce hallucinations of LLMs, whether such citations can themselves be considered trustworthy is unclear (Peskoff & Stewart, 2023; Zuccon et al., 2023; Gravel et al., 2023). In fact, previous work has explored generative search engines like Bing Chat, NeevaAI, perplexity.ai, and YouChat that provide inline citations, and find that they frequently contain unsupported statements and inaccurate citations (Liu et al., 2023).

**Feature Attribution-Based Explanations** The field of neural network interpretability has put forth a huge number of techniques that aim to measure the importance or relevance of each input feature on a model's prediction. Perturbation-based techniques including leave-one-out and masking have been applied on features like words and phrases (Li et al., 2016), input word-vector dimensions, intermediate hidden units (Li et al., 2017), and token spans (Wu et al., 2021). The gradient of a model's output with respect to an input feature indicates the sensitivity of output to input changes and has therefore been used to compute attribution scores (Yin & Neubig, 2022; Mohebbi et al., 2021; Sanyal & Ren, 2021; Sikdar et al., 2021; Enguehard, 2023). We adapted this simple gradient-based attribution approach as one of our baselines. Since attention computes an adaptively weighted average of activations, attention weights can be seen as capturing the correlation between inputs. Attention weights have therefore also been a popular tool to identify important information to the model (Pruthi et al., 2020a; Serrano & Smith, 2019; Wiegreffe & Pinter, 2019) and we therefore use attention weight as one of our baselines. Local interpretable surrogate models are also developed to interpret model outputs (Ribeiro et al., 2016; Lundberg & Lee, 2017).

**Attributing Predictions to Training Data** Apart from attributing an LLM's predictions to input features that correspond to elements in its context, it can be of practical interest to uncover the training data that influenced a particular prediction. Such is the goal of influence functions (Koh & Liang, 2020), which estimate the impact of removing a particular training example by computing the alignment of gradients through a bilinear form with the inverse Hessian. The high cost of computing and inverting the Hessian, as well as the cost of computing gradients for every in example within a potentially gargantuan training dataset, has led to a great deal of work on computing influence functions more efficiently, particularly for LLMs (Grosse et al., 2023; Choe et al., 2024; Pruthi et al., 2020b). Apart from costs, the reliability of influence functions has been questioned (Basu et al., 2021; Bae et al., 2022). As exact LOO for dataset attribution requires a full training run for each data point, most training data attribution methods resort to evaluating with proxy metrics like task accuracy as a function of the number of high-attribution training examples removed, instead of ground truth LOO attributions as in our work. Beyond directly measuring influence, other works have sought to understand the influence of pre-training data on factual knowledge internalized by the model (Kandpal et al., 2023; Chang et al., 2024; Razeghi et al., 2022; Antoniades et al., 2024). More broadly, the impact of pre-training data on LLM behavior and performance is an important notion for the well-studied problem of data selection; see (Albalak et al., 2024).

## 6 CONCLUSION

In this paper, we introduced AttriBoT, a **B**ag **o**f **T**ricks for efficiently attributing an LLM's predictions to spans of text in its input context. AttiBoT leverages a series of novel techniques for approximating the attributions of a large target model, including reusing computations through key-value caching, performing hierarchical attributions to reduce the number of forward passes, and leveraging a smaller proxy model whose attributions reliably approximate those of the target model. Taken together, AttriBoT can provide a $>300\times$ speedup, making attributing a response $30\times$ more efficient than even generating a response in realistic settings, while remaining more faithful to the target model's attributions than prior methods for efficient context attribution. In addition, the components of AttriBoT can be included, excluded, and tuned to produce a Pareto-optimal trade-off between faithfulness and speed. While our experimental results primarily focus on the general setting of open-book question answering, we anticipate AttriBoT will also be useful for detecting malicious prompts and model hallucinations. We hope that our efficient and easy-to-use implementation of AttriBoT[1] ensures that it has real-world impact and also enables future work on efficient context attribution.

## 7 ACKNOWLEDGEMENTS

We thank Haokun Liu, Brian Lester, Wanru Zhao, and the members of Vector Institute for their valuable feedback and support. Resources used in preparing this research were provided, in part, by the Province of Ontario, the Government of Canada through CIFAR, and companies sponsoring the Vector Institute https://vectorinstitute.ai/partnerships/current-partners/.

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

# A    THEORETICAL EFFICIENCY DERIVATIONS

In this section, we provide derivations for the theoretical efficiencies reported in Table 1.

**Preliminaries**    We assume a target model with $P$ parameters, a context $C$ containing $|C|$ sources, and that each source in the context contains exactly $T$ tokens. Following Kaplan et al. (2020) and Hoffmann et al. (2022), we approximate the number of FLOPs for a forward pass of a $P$-parameter model on a sequence of $T$ tokens to be $2PT$ FLOPs.

To simplify the analysis, we ignore the fact the query and response are also part of the model's input, as these are typically negligible compared to the length of the context. Accounting for the query and response would simply add a small constant factor to each method's theoretical runtime. Finally, we assume that the likelihood of the full sequence under the target model, $p_\theta(R|Q, C)$ can be computed while generating $R$, and is thus not accounted for in the analysis of each attribution method.

## A.1    EXACT LOO ATTRIBUTION

Exactly calculating LOO attributions under the target model requires computing $p_\theta(R|Q, C \setminus \{s_i\}$ for all in $i \in 1, \ldots, |C|$. In total, this is $|C|$ forward passes each on sequences containing $T(C-1)$ tokens. Thus, the number of FLOPs needed to compute attributions is $2PT|C|(|C|-1)$.

## A.2    KV CACHING

KV caching allows us to avoid FLOPs spent on prefixes for which keys and values have already been computed. Since keys and values can be cached for the full sequence while computing $p_\theta(R|Q, C)$, we can avoid recomputing the keys and values for the first $i-1$ sources while computing $p_\theta(R|Q, C \setminus \{s_i\})$. Thus the total number of FLOPs for computing all LOO attributions can be written as:

$$\sum_{i=1}^{|C|} 2PT(|C|-i) = PT|C|(|C|-1)$$

This provides a speedup of a factor of 2 compared to computing LOO attributions with no caching.

## A.3    PROXY MODELING

Proxy modeling is identical to computing exact LOO attributions, except with a model with fewer parameters. If the number of proxy model parameters is $P'$, then the number of FLOPs needed to compute attributions is $2P'T|C|(|C|-1)$ and the speedup gained by using a proxy model is $P/P'$.

## A.4    HIERARCHICAL ATTRIBUTION

Our analysis of hierarchical attribution makes the simplifying assumption that each source group in the context contains exactly $H$ sources.

We start by first considering the initial source group-scoring step in the hierarchical attribution algorithm. This step requires performing $|C|/H$ forward passes (one for each source group) each on a sequence of $T(|C| - H)$ tokens. Thus, the initial source group scoring step uses $2PT(|C|/H)(|C| - H)$ FLOPs.

The second step in hierarchical attribution computes scores at the source level on a shortened context containing only $\beta|C|$ sources. Thus, the second step of hierarchical attribution uses $2PT(\beta|C|)(\beta|C| - 1)$.

Combining these steps together, we can compute the total number of FLOPs for hierarchical attribution:

$$\begin{aligned}
\text{Total FLOPs} &= 2PT\left(\frac{|C|}{H}\right)(|C| - H) + 2PT(\beta|C|)(\beta|C| - 1) \\
&= 2PTC\left(\frac{|C|}{H} - 1 + \beta^2|C| - \beta\right) \\
&= 2PTC\left(\left(\beta^2 + \frac{1}{H}\right)|C| - \beta - 1\right)
\end{aligned}$$

By comparing to the number of FLOPs used for exact LOO attribution we get the following speedup factor:

$$\frac{|C| - 1}{\left(\beta^2 + \frac{1}{H}\right)|C| - \beta - 1} = \frac{H(|C| - 1)}{(\beta^2 H + 1)|C| - \beta H - H}$$

Next, we consider the speedup as the number of contexts grows and the number of source groups kept after the first group-level scoring pass is kept constant, i.e., $\beta = k/|C|$ for some constant $k$.

$$\begin{aligned}
\frac{H(|C| - 1)}{(\beta^2 H + 1)|C| - \beta H - H} &= \frac{H(|C| - 1)}{\left(\left(\frac{k}{|C|}\right)^2 H + 1\right)|C| - \frac{k}{|C|}H - H} \\
&= \frac{H(|C| - 1)}{\frac{k^2}{|C|}H + |C| - \frac{k}{|C|}H - H} \\
&\approx \frac{H|C|}{|C|} \quad \text{for large } |C| \\
&= H
\end{aligned}$$

Thus, for contexts with many sources, hierarchical attribution's speedup approaches $H$, the number of sources per source group.

## A.5  PROXY MODEL PRUNING

The first stage of proxy model pruning simply uses a proxy model to score each source. Using the result from Appendix A.3, this operation uses $2P'T|C|(|C| - 1)$ FLOPs, where $P'$ is the number of parameters in the proxy model. The second stage of the proxy model pruning algorithm uses the target model to score the remaining $\alpha|C|$ sources, requiring $2PT(\beta|C|)(\beta|C| - 1)$ FLOPs. Thus the total number of FLOPs can be written as:

$$\begin{aligned}
\text{Total FLOPs} &= 2P'T|C|(|C| - 1) + 2PT(\beta|C|)(\alpha|C| - 1) \\
&= 2PT|C|\left(\left(\frac{P'}{P}\right)(|C| - 1) + \alpha(\alpha|C| - 1)\right) \\
&= 2PT|C|\left(\frac{P'}{P}|C| - \frac{P'}{P} + \alpha^2|C| - \alpha\right) \\
&= 2PT|C|\left(\left(\alpha^2 + \frac{P'}{P}\right)|C| - \alpha - \frac{P'}{P}\right)
\end{aligned}$$

When compared to exact LOO attribution with the target model, this method provides a speedup of:

$$\frac{|C| - 1}{\left(\alpha^2 + \frac{P'}{P}\right)|C| - \alpha - \frac{P'}{P}} = \frac{P(|C| - 1)}{(\alpha^2 P + P')|C| - \alpha P - P'}$$

Next, we consider the speedup as the number of contexts grows and the number of sources kept after proxy model pruning is kept constant, i.e. $\alpha = k/|C|$ for some constant $k$.

$$
\frac{P(|C| - 1)}{(\alpha^2 P + P')|C| - \alpha P - P'} = \frac{P(|C| - 1)}{\left(\left(\frac{k}{|C|}\right)^2 P + P'\right)|C| - \frac{k}{|C|}P - P'}
$$

$$
= \frac{P(|C| - 1)}{\frac{k^2}{|C|}P + P'|C| - \frac{k}{|C|}P - P'}
$$

$$
\approx \frac{P|C|}{P} \quad \text{for large } |C|
$$

$$
= P
$$

Thus, as the number of sources in the context grows, the speedup of proxy model pruning approaches $P/P'$.

## B  FURTHER DETAILS ON EXPERIMENTAL SETUP

In this section, we provide more details on the experimental setup we use to evaluate context attribution methods.

### B.1  DATASETS

**Preprocessing**  Before evaluating context attribution methods on SQuAD 2.0, HotpotQA, and QASPER, we first take the following preprocessing steps to make them suitable for running context attribution.

SQuAD2.0:

1. Filter dataset to only include examples that are labeled as possible to answer given the context (SQuAD 2.0 contains both questions where the answer is and is not present in the contextual information).

2. Truncate each example's supporting Wikipedia page context to 10 paragraphs and filter dataset to only examples where the answer lies in the first 10 paragraphs of the context. This is meant to filter out examples with a very large number of context sources, as computing exact LOO attributions on these examples is impractical.

3. Randomly sample 1000 examples from the remaining set of examples.

Hotpot QA:

1. Sample the first 1000 examples of the dataset.

QASPER:

1. Filter dataset to only examples with contexts containing fewer than 60 paragraphs. This limits the context length to make exact LOO attribution tractable and also removes improperly parsed contexts where every line in a LaTeX math environment is listed as its own paragraph.

2. Randomly sample 1000 examples from the remaining set of examples. In our experiment, there are 2 empty examples and they are ignored, resulting in 998 examples.

We report summary statistics for each dataset after preprocessing in Table 2.

**Prompting**  For each dataset, we use the following prompt templates to format a query and context into a prompt suitable for instruction-tuned models.

SQuAD 2.0 and HotpotQA:

```
Answer the question based on the provided context
```

| Dataset | Examples | Source Type | Sources/ Example | Source Groups/ Example | Tokens/ Example | Generation GPU-seconds/Example |
|---|---|---|---|---|---|---|
| HotpotQA | 1000 | Sentence | 41.2 | 9.9 | 1,249 | 18 |
| SQuAD2.0 | 1000 | Sentence | 50.1 | 10.0 | 1,592 | 19.6 |
| QASPER | 998 | Paragraph | 38.2 | 12.7 | 4,116 | 51.7 |

Table 2: Summary statistics for each of the evaluation datasets.

```
Context:
```
```
{context}
```
```
Question:  {question}
```

QASPER:
```
Answer the question based on the following scientific paper:
```
```
{context}
```
```
Question:  {question}
```

## B.2 OUTLIER DETECTION WITH EXTREME STUDENTIZED DEVIATE TEST

The generalized Extreme Studentized Deviate (ESD) test (Rosner, 1983) is a statistical test for detecting an unknown number of outliers in a univariate set of data. This test is an iterated form of Grubbs' test for detecting a single outlier in a set of data (Grubbs, 1969).

The ESD test is parameterized by the maximum possible number of outliers, $k$, and the significance value $\alpha$. It works by iteratively selecting the largest value in the dataset, computing that value's Grubbs' statistic $G$, testing if $G$ is larger than the critical value $G_{crit}$, and removing the tested value from the dataset. This process stops when the selected value no longer has a Grubbs' statistic greater than the critical value or $k$ values are tested.

The Grubbs' statistic and critical value are defined as is defined as follows for a collection of data $Y_1, \ldots, Y_N$:

$$G = \frac{\max_{i=1,\ldots,N} Y_i - \bar{Y}}{s}$$
$$G_{crit} = \frac{(N-1) \times t}{\sqrt{N} \times \sqrt{N-2+t^2}}$$

where $\bar{Y}$ and $s$ are the mean and standard deviation of $Y_1, \ldots, Y_N$, and $t$ is the upper critical value of the t-distribution with $N-2$ degrees of freedom and a significance level of $\frac{\alpha}{2n}$.

For our experiments we set the significance level $\alpha = 0.05$ and the maximum number of outliers $k = 50$.

## B.3 FURTHER DETAILS ON CONTEXTCITE BASELINE

ContextCite works by training a linear surrogate model to approximate the LOO attributions of a target model. Given context sources $S = \{s_1, \ldots, s_{|C|}\}$, ContextCite samples $n$ ablation vectors $\vec{v}_1, \ldots, \vec{v}_n \in \mathbb{R}^{|C|} \sim \text{Bernoulli}(p)$ that act as masks indicating which context sources to remove/keep for each forward pass. Based on these ablation vectors, ContextCite performs a single forward pass on each ablated context and records the change in logit-scaled response probability, $\tau_i$, caused by the removal of the ablated sources. Next, a Lasso regression surrogate model is fit to map each ablation vector $\vec{v}_i$ to the corresponding change in response probability $\tau_i$. Finally, the attribution scores for context source $j$ is simply the weight that the trained Lasso model assigns to the $j$'th source.

# C    ADDITIONAL EXPERIMENTAL RESULTS

## C.1    EFFICIENCY VS. ACCURACY TRADEOFF FOR ATTRIBOT METHODS

We provide plots illustrating the efficiency vs. accuracy tradeoff for each AttriBoT acceleration method in Figure 3, Figure 4, and Figure 5. Overall, we find that each method provides parameters that effectively vary the accuracy-efficiency tradeoff and are Pareto optimal with respect to baseline context attribution methods.

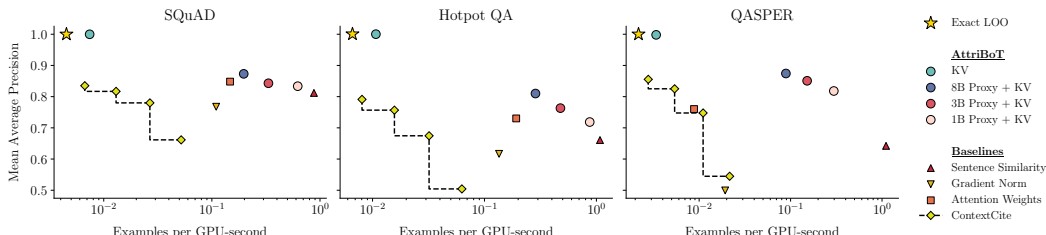

Figure 3: Plot showing accuracy vs. efficiency tradeoff of proxy modeling. We find that smaller proxy models produce attributions that are less faithful to the target model's attributions.

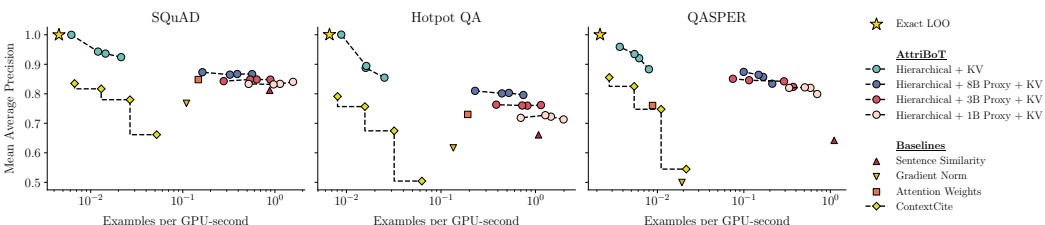

Figure 4: Plot showing accuracy vs. efficiency tradeoff of hierarchical attribution with and without the use of a proxy model. Both varying the size of the proxy model and tuning the number of source groups to retain, $\beta$ effectively trades attribution faithfulness for speed.

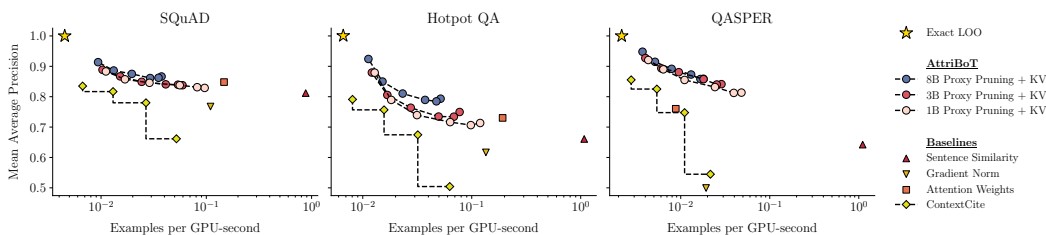

Figure 5: Plot showing the accuracy vs. efficiency tradeoff of proxy model pruning. We find that varying the size of the proxy model and the fraction of sources to retain, $\alpha$ effectively trades attribution faithfulness for speed.

## C.2    ATTRIBOT'S EFFECTIVENESS ACROSS DIFFERENT TARGET MODELS

To demonstrate AttriBoT's effectiveness for approximating LOO attributions for different target models, we replicate the experiments on HotpotQA using Qwen 72B as the target model. Shown in Figure 6, we observe a nearly identical Pareto front as seen in the Llama model family. Aside from the attention weights baseline, AttriBoT is Pareto optimal at all points on the Pareto front. We hypothesize that the lack of Pareto optimality compared to the attention weights baseline comes from surprisingly good performance of the baseline method compared to its performance in other experiments, achieving a mAP about 10 points higher than when used with Llama 70B.

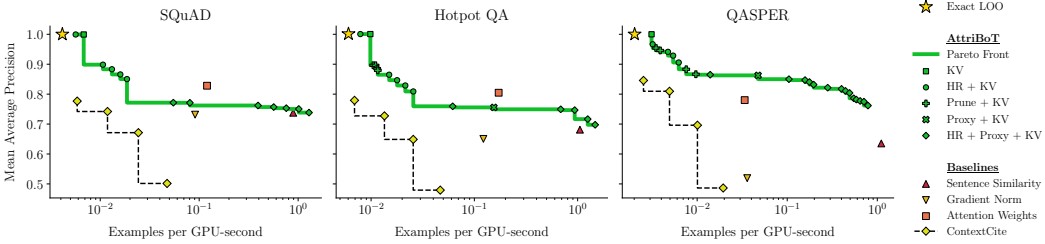

Figure 6: We replicate the experiments on SQuAD, HotpotQA and QASPER using Qwen 72B as the target model to demonstrate the AttriBoT's effectiveness across different target models.

## C.3 AFFECT OF TRAINING DATA DISTRIBUTION OF PROXY MODEL

To further investigate the potential impact of proxy model selection and their training data distribution, we considered mismatched proxy models including Mistral 7B Instruct v0.3, as shown in Figure 7, Figure 8, Figure 9, and Figure 10. While our bag of tricks are still effective, using a mismatched proxy model can degrade performance somewhat compared with proxy models trained with same data or distilled from the larger target models.

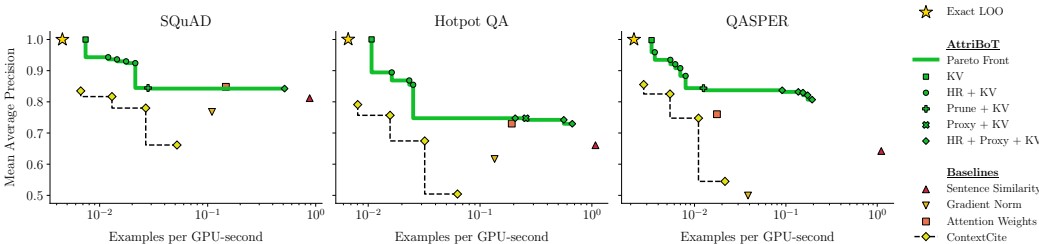

Figure 7: Using Llama 3.1 70B Instruct as the target model and Mistral 7B Instruct v0.3 as the proxy model.

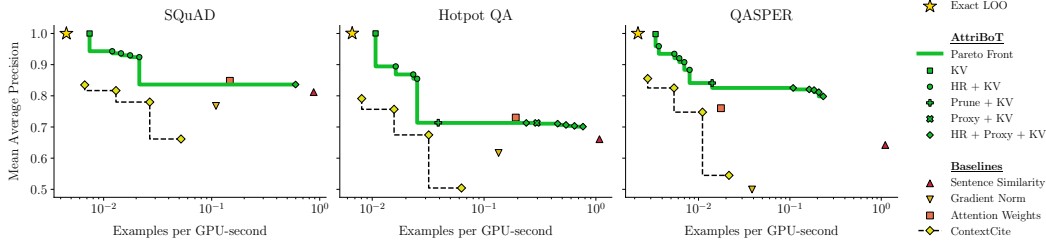

Figure 8: Using Llama 3.1 70B Instruct as the target model and Qwen 2.5 7B Instruct as the proxy model.

## C.4 MEAN AVERAGE PRECISION COMPARING WITH HUMAN ANNOTATIONS

The HotpotQA dataset comes with human-annotated "important" text spans in each example's context. While the primary purpose of AttriBoT is to approximate a target model's LOO attributions, here we evaluate the agreement between AttriBoT attributions and human-annotated gold sources. We find that AttriBoT's context attributions closely align with human annotations, achieving only a 10% drop in mAP compared to exact LOO attributions, while achieving a 300× speedup. Results are shown for multiple target models in Figure 11.

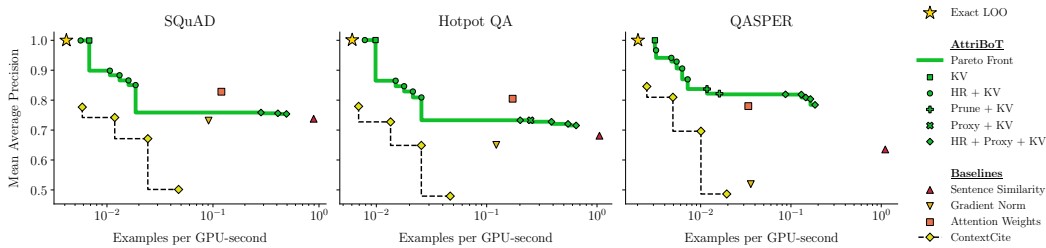

Figure 9: Using Qwen 2.5 72B Instruct as the target model and Mistral 7B Instruct v0.3 as the proxy model.

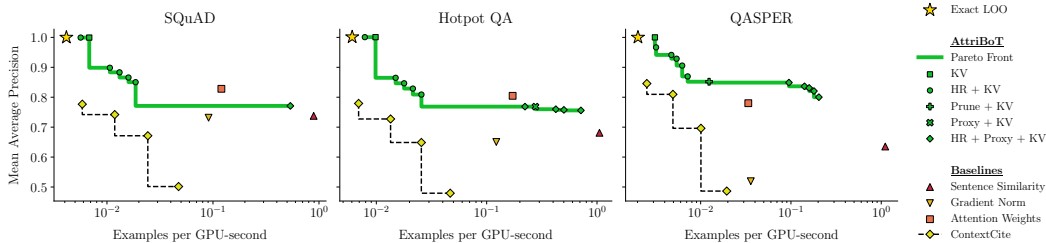

Figure 10: Using Qwen 2.5 72B Instruct as the target model and Llama 3.1 8B Instruct as the proxy model.

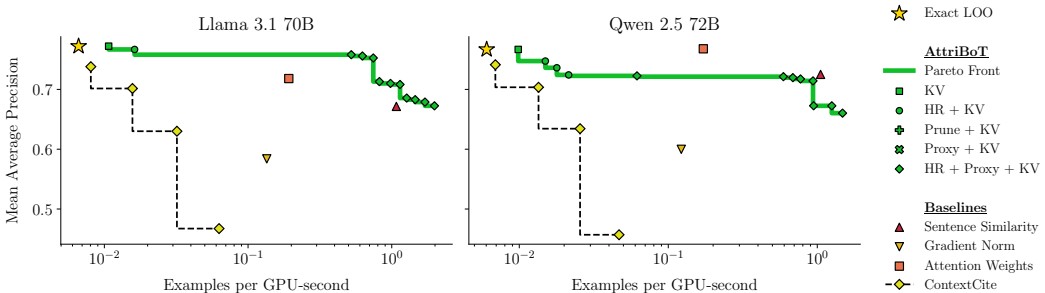

Figure 11: Mean average precision comparing with human annotation on HotpotQA dataset with Llama 3 family and Qwen 2.5 family.

# D   PRACTITIONER'S GUIDE TO USING ATTRIBOT

## D.1   METHOD AND HYPERPARAMETER SELECTION

KV caching should always be employed because is effectively lossless and can be combined with other methods. Whenever feasible, float16 should be used to avoid possible floating point errors. To further reduce LOO costs, we recommend starting with hierarchical attribution and pruning. For tasks like open-domain QA where only a small subset of the context is relevant, retaining 3 to 5 paragraphs or sentences is generally sufficient to maintain faithfulness while achieving significant acceleration. For further speedup, consider using smaller models within the same model family or distilled versions of the target model. Models in the same family with sizes as small as 1B or 0.5B parameters can remain faithful even for complex tasks. Additionally, combining proxy methods with hierarchical attribution is advisable, as their performance degradation does not compound linearly.

## D.2   INPUT PREPROCESSING

In settings such as RAG and in-context learning, input information may be duplicated due to the retrieval process. Such duplication poses challenges for LOO context attribution or any attribution

method that aims to approximate the LOO error, as removing one instance of that information would likely not change the model's response likelihood. We strongly recommend deduplicating input before performing LOO context attribution.

In scenarios where input information is unstructured, we suggest chunking the input by parsing sentences and grouping a few (e.g., 10) sentences together to facilitate hierarchical context attribution.

