# OpenReview forum: "AttriBoT: A Bag of Tricks for Efficiently Approximating Leave-One-Out Context Attribution"
_ICLR.cc/2025/Conference — ICLR 2025 Poster_

### Official Review · Reviewer_Y8fS · 2024-11-02

**Soundness:** 3
**Presentation:** 3
**Contribution:** 3
**Rating:** 6
**Confidence:** 3

**Summary:**

This paper focuses on efficiently approximating the LOO error for context attribution. Based on key observations, the author develops AttriBoT, which utilizes cached activations, hierarchical attribution, and smaller proxy models to facilitate large-scale attribution computation.

**Strengths:**

The issue addressed in this paper is significant and holds substantial research value. Furthermore, the writing is clear, and the arguments are easily understandable, with an intuitive methodology.

**Weaknesses:**

See questions part for more details.

**Questions:**

1. My understanding is that the Pareto-optimal conclusion regarding efficiency and performance is derived from Figure 2. While consistent results are observed across the three datasets, does the Pareto-optimal outcome have corresponding theoretical support?

2. In Section 4.1.1, the authors introduce two target models, the Llama3.1 series and the Qwen 2.5 series, for evaluating the AttriBoT method. Can this method be applied to closed-source large models? Additionally, is it applicable to LLMs with architectures other than Transformers?

3. As a general approach, the authors mention the broad applications of context attribution, but only use OBQA for data. Can the effectiveness of the proposed method be further validated in other contexts or task types?

---

> ### Author Response · Authors · 2024-11-22
> **Author Response**
>
> Thank you for taking the time to thoughtfully review our work. Below, we provide some further context and try to clarify the questions identified in your review.
>
> # Theoretical Support of Pareto-Optimality
> We are not able to make theoretical claims that AttriBoT is Pareto-optimal in general because this finding involves comparison to methods that work quite differently than LOO attribution (e.g. attention-based attribution).
>
> # Closed Source Language Models
> The only assumption made by our methods (and LOO attribution in general) is that we can compute the conditional likelihood of a response given a context. Thus, our method can be applied to any model that provides likelihoods as part of its API. Whether AttriBoT could be applied to a closed-source LLM therefore depends on whether the LLM's API provides likelihood values.
>
> # Non-Transformer Architectures
> In principle, the hierarchical, proxy modeling, and pruning methods within AttriBoT could be applied to non-Transformer language models. KV caching would only extend to other architectures if, as in causal self-attention, it is valid to reuse past KV states.
>
> # Evaluation Outside of Open-Book QA
> We focus on open-book QA in our work because it is realistic and representative of many typical LLM use-cases (i.e. any query that involves conditioning on specific spans within a larger context) while being straightforward to evaluate. Other LLM use-cases, such as abstractive summarization, most or many context spans might be informative, so measuring the LOO error might not provide useful or actionable insights. If you have specific benchmarks/settings that you would like us to include, please let us know and we will do our best to add them.

---

> ### Comment · Reviewer_Y8fS · 2024-11-25
> **Reviewer Feedback on Author's Revisions**
>
> Thank you for your response to the issues raised.

---

### Official Review · Reviewer_5ZgU · 2024-11-03

**Soundness:** 3
**Presentation:** 3
**Contribution:** 3
**Rating:** 6
**Confidence:** 4

**Summary:**

The authors present a method leave-one out for contextual attribution, i.e. how does a certain part in a model prompt influence the output of the model. Leave-one-out has been mainly impractical due to computational cost and the authors address efficiency via
1. KV caching
2. hierarchical evaluation
3. approximation via smaller (proxy) models

The authors demonstrate that this combination can produce satisfying results with up to 300x speedup, of 30x faster than generating the response (with the bigger model alone).

**Strengths:**

- Context attribution is an important problem, including but not limited to the areas the authors called out in the paper. For example interpretability, reliability, safety, privacy/confidentiality, etc.
- Solid discussion of computational cost, and gains by introducing the individual optimizations
- I like that authors combine and discuss optimizations across the stack ranging from hybrid modelling by combining models of different complexity, leveraging insights of the problem (hierarchical approach), down to lower level optimization such as KV caching

**Weaknesses:**

1. I do not consider the way KV caching is leveraged in this paper as a novelty. KV caching over multiple requests comes for free with well established inferencing frameworks available. However KUDOs to the authors that they discuss the importance of KV cachine in detail, including it's cost saving.
2. It's an important, but fairly narrow area

Some minor concerns:
1. How well would this generalize to more complex reasoning tasks. Recent literature suggests that reasoning capabilities in models vary quite considerable, and only the largest models can properly come up with reasoning solutions, or even judge the output beyond style or simpler tasks (if properly assessed). How well does the proxy model approach generalize if there is a larger gap in model capabilities, and for more complex tasks.
2. How well does the hierarchical approach work in real world scenarios, where information can appear in multiple places (also see in questions section)

**Questions:**

I can see how hierarchical attribution works if context only appears in a single, localized place int he prompt. However, with most real world RAG approaches, we would expect information to be present & spread accross multiple places in the document. What would happen in such a case?

---

> ### Author Response · Authors · 2024-11-22
> **Author Response**
>
> Thank you for taking the time to thoughtfully review our work. Below, we provide some further context and discuss the questions and weaknesses identified in your review.
>
> # KV Caching
> We agree with the reviewer, that KV caching comes at little implementation cost. However, we do note that our use of KV caching is notable and novel in this context for the following reasons:
> 1. Standard KV caching (where the entire context is cached and re-used as-is) is not applicable to LOO attribution since the context changes for each pass of leave-one-out. AttriBoT's  novel application of KV caching involves identifying the longest possible pre-computed prefix that can be used based on the current context and past contexts, with additional complications when processing a context hierarchically.
> 2. KV caching does not provide a speedup by default for all context attribution methods that involve multiple forward passes. An example of this is ContextCite, which performs each forward pass on a context with a randomly selected set of sources removed. Since the source removal is probabilistic, contexts on average have very short prefixes in common resulting in virtually zero performance gain from KV caching.
>
> # Proxy Modeling for Complex Tasks
> This is an excellent point that certain complex tasks are only possible for large models, thus potentially breaking the proxy modeling assumption. This is seen in our proxy modeling results (Figure 3) where larger proxy models yield better approximate LOO attributions than smaller proxy models. One important caveat to note is that for proxy modeling to work, it is *not* necessary that the proxy model be able to perform the task on its own. This is because when we approximate the LOO attributions for a large target model using a small proxy model, we use the response generated by the target model and evaluate this response’s likelihood (given different contexts) using the proxy model. Thus, the proxy does not need to produce the same response as the target model, but rather must emulate the conditional likelihood of a response given a context. We do note that we include results on HotpotQA, which involves reasoning over multiple "hops" to answer questions. We have added corresponding clarifications to the paper.
>
> # Duplicate Context Information
> It is a valid point that duplicate contextual information can cause problems for the methods we propose. If the same piece of information appears multiple multiple times in the context, then simply removing one instance of that information would likely not change the model’s response likelihood. However, this is not unique to our methods, but rather to any attribution method that aims to approximate the LOO error. For instance, many popular training data attribution methods, such as influence functions, also suffer from this issue since they approximate LOO error for training data. In our newly added ‘Practitioner’s Guide to AttriBoT’ in Appendix D, we suggest users to deduplicate input before performing LOO attribution.
>
> An alternative approach for situations with duplicate context information would be to compute attributions based on notions of value other than the LOO error – such as the Shapley value. In general, however, Shapley values are much more expensive to compute than the LOO error, as they require performing an exponential (in the number of context sources) number of forward passes compared to a linear number of forward passes needed for the LOO error.

---

### Official Review · Reviewer_uZjG · 2024-11-03

**Soundness:** 2
**Presentation:** 2
**Contribution:** 2
**Rating:** 6
**Confidence:** 2

**Summary:**

This paper introduces AttriBoT, a novel collection of techniques for efficiently approximating leave-one-out (LOO) context attribution in large language models. The authors present three key insights: 1) caching attention key-value pairs can avoid redundant computations, 2) hierarchical attribution can reduce necessary computations through pruning, and 3) smaller proxy models can effectively approximate larger target models' attributions. The combined approach achieves remarkable efficiency gains, providing up to 300x speedup while maintaining high fidelity to the target model's attributions compared to baselines. The method is extensively evaluated across different model scales and datasets, demonstrating consistent performance improvements in open-book question answering tasks.

**Strengths:**

1. Technical Innovation & Practicality
- Introduces multiple complementary techniques that can be used independently or combined
- Provides both theoretical and empirical justification for each component
- Achieves practical speedups that make attribution feasible in real-world applications
- Releases efficient implementation to benefit the research community

2. Comprehensive Evaluation
- Tests across multiple datasets (SQuAD, HotpotQA, QASPER)
- Evaluates with different model families (Llama, Qwen)
- Includes thorough ablation studies of different components
- Provides detailed theoretical efficiency analysis with derivations

3. Strong Empirical Results
- Demonstrates clear Pareto-optimal trade-offs between speed and accuracy
- Shows consistent performance across different context lengths and model sizes
- Achieves significant speedups (300x) while maintaining high attribution fidelity
- Outperforms existing attribution methods like ContextCite

4. Clear and Rigorous Methodology
- Well-defined metrics for measuring attribution quality
- Thorough baselines for comparison
- Careful experimental design with appropriate controls
- Detailed implementation specifications

**Weaknesses:**

1. Limited Scope of Applications
- Primary evaluation focuses on open-book QA tasks
- Could explore effectiveness in other applications like detecting malicious prompts or hallucinations
- Could demonstrate utility for real-time attribution scenarios

2. Parameter Sensitivity Analysis
- Could provide more guidance on selecting optimal parameters (e.g., pruning thresholds)
- Limited discussion of how parameter choices might vary across different use cases
- Could explore automatic parameter tuning approaches

**Questions:**

None

---

> ### Author Response · Authors · 2024-11-22
> **Author Response**
>
> Thank you for taking the time to thoughtfully review our work. Below, we provide some further context and discuss the weaknesses identified in your review.
>
> # Evaluation Other Than Open-Book QA
> We focus on open-book QA in our work because it is realistic and representative of many typical LLM use-cases (i.e. any query that involves conditioning on specific spans within a larger context) while being straightforward to evaluate. Other LLM use-cases, such as abstractive summarization, most or many context spans might be informative, so measuring the LOO error might not provide useful or actionable insights. If you have specific benchmarks/settings that you would like us to include, please let us know and we will do our best to add them.
>
> # Hyperparameter Selection
> We agree that with a set of methods like AttriBoT, that can be mixed and matched to achieve different efficiency vs. accuracy tradeoffs, selecting an algorithm combination and associated hyperparameters can be difficult. To mitigate this we do two things:
>
> 1. We amend Figure 2 to show which algorithm combinations are represented by different points on the AttriBoT Pareto front. This gives readers better insight into which combinations of methods tend to provide higher accuracy and lower efficiency vs. lower accuracy and higher efficiency.
> 2. We have added a “Practitioner’s Guide to AttriBoT” as Appendix D, in which we summarize the relative efficiency of different algorithm combinations, provide insight into hyperparameter (e.g., hierarchical thresholds) selection based on our experiments, and advise users on best practices for data preprocessing before performing context attribution.

---

### Official Review · Reviewer_f2FJ · 2024-11-04

**Soundness:** 3
**Presentation:** 3
**Contribution:** 2
**Rating:** 6
**Confidence:** 3

**Summary:**

This paper studies how to perform leave-one-out context attribution with resource limitations. For Large language models, performing leave-one-out context is very expensive. Therefore, this paper proposes AttriBot, including the following key points:

- Key-value caching
- Hierarchical attribution
- Proxy modeling

This paper shows that their method achieves lots of speedup comparing to the scenarios where their method is not used while maintaining the performance.

**Strengths:**

- The proposed methods are simple but effective.
- This research direction is useful.
- All findings are supported by experiments (e.g., smaller model in the same model family approximate the target model’s LOO attribution score well)

**Weaknesses:**

- [Major] The proposed methods are too simple and not novel, the messages are not surprising. In fact, I am not sure whether key-value caching can be considered as a contribution here or not. This is just one general trick for speeding up LLM’s inference. The other twos are slightly more novel, but still pretty straightforward. Therefore, to mitigate this weakness, I think this paper requires more extensive experiments to have sufficient technical contributions.
- [Major] More experiments are needed. For instance, this paper only considers Llama model family and Qwen family, and the experiments of many important findings seems only conducted on LLaMA (Figure 1, please correct me if I am wrong, I also checked appendix).
- [Medium] More findings are needed. Just one random example — for other models, whether we observe similar phenomenon as shown on LLaMA model family? If not, what could be the reason? Training dataset? Or other thing? At least, there are still many interesting questions to answer. I think if the authors can dive deeper in this directions, this paper can be a good paper. But now the technical contributions are not sufficient yet.

**Questions:**

See the weakness section.

**Details Of Ethics Concerns:**

No.

---

> ### Author Response · Authors · 2024-11-22
> **Author Response**
>
> Thank you for taking the time to thoroughly read our work. Below, we discuss some of the concerns brought up in your review.
>
> # Simple Methods
>
> Regarding KV caching, while KV caching is a general trick for speeding up inference, we note that standard KV caching (where the entire context is cached and re-used as-is) is not applicable to LOO attribution since the context changes for each pass of leave-one-out. While we agree that adapting KV caching to LOO attribution is not a fundamentally new method, we argue that it does involve some novelty. Furthermore, we would like to note that KV caching actually cannot be applied out of the box to any attribution method that performs repeated inference. For instance, ContextCite gains almost no improvement in performance from KV caching, since for every forward pass a randomly sampled set of sources are removed from the context resulting in a low probability of prefix-sharing across forward passes.
>
> More broadly, we agree that the methods proposed in this work are quite straightforward. However, seeing as how these methods actually work quite well for efficiently approximating LOO attributions, we argue that this should not be considered a weakness. Especially in newer research areas, like context attribution, it is important to start with simple methods to understand areas of improvement that actually require complex solutions.
>
> Furthermore, we make explicit the assumptions of each method (e.g., sum of k leave-one-out attributions is approximately equal to leave-k-out attribution, proxy model attributions approximate target model attributions when both models are attributing the same response, etc.) and provide evidence that these assumptions hold in realistic settings.
>
> # Experiments on Multiple Models and additional findings
> We agree with this point. Our main experiments used the Llama model family and we also reported results on the HotPotQA dataset using the Qwen model family to show that these results generalize across models. Since receiving your feedback we have replicated all of the experiments (QASPER, SQuAD, and HotPotQA) using the Qwen model family (Figure 6) and see nearly identical results, demonstrating the generalizability of our method. We also included new results considering mismatched proxy models including Mistral 7B (Figures 7-10) and find that, while still effective, using a mismatched proxy model can degrade performance somewhat.

---

> > ### Comment · Reviewer_f2FJ · 2024-11-26
> >
> > Thanks so much for your thoughtful response. I will increase my score to 6.

---

### Official Review · Reviewer_zBXc · 2024-11-04

**Soundness:** 3
**Presentation:** 3
**Contribution:** 3
**Rating:** 8
**Confidence:** 4

**Summary:**

The paper introduces AttriBoT, a set of methods for efficiently approximating leave-one-out context attribution at LLM scale. In particular, the authors show that (1) using context caching reduces the number of FLOPs approximately twice compared to naive computation of LOO; (2) Hierarchical Leave-k-out attribution efficiently approximates leave-one-out attribution; and (3) LOO attribution scores for larger models can be approximated by LOO attribution scores for smaller models from the same family. The authors demonstrate that combining these techniques can achieve a 300x speedup in the computation of approximate LOO attribution scores, with only a 10% drop in mean average precision.

**Strengths:**

1. The paper addresses an important problem of context attribution. The leave-one-out (LOO) error method for context attribution is computationally expensive due to the need to re-generate the text after removing each piece from the context. The paper proposes a set of techniques for approximating the LOO scores while significantly reducing computational cost. The proposed methods are very intuitive and can be easily applied in practice.

2. The paper evaluates the proposed combination of approximation and computational techniques across a number of datasets and language models. The authors also compare AttriBoT against several baselines and demonstrate the optimality of AttriBoT's precision-speed Pareto front compared to other context attribution methods.

**Weaknesses:**

While most of the proposed techniques are not entirely novel in the field—for example, KV caching has been applied to other problems, as has transferability between language models from the same family—to my knowledge, this is the first paper where these approaches have been applied to the context attribution problem.

**Questions:**

1. Could the authors please label the AttriBoT combinations in Figure 2 for better clarity?

2. It might be beneficial to include more model families in the empirical experiments, such as Gemma.

---

> ### Author Response · Authors · 2024-11-22
> **Author Response**
>
> Thank you for taking the time to engage with our paper. Regarding your questions:
>
> 1. We have amended Figure 2 in our submission to identify which points on the Pareto front correspond to different methods. Across most experiments, the rough ordering of methods on the Pareto front from most efficient to least efficient is as follows: Hierarchical + Proxy + KV Caching, Proxy + KV Caching, Hierarchical + KV Caching, Pruning + KV Caching, and KV Caching.
>
> 2. Our original experiments all used the Llama model family except for one experiment on HotPotQA that we replicated using the Qwen 2.5 model family. The intention of this Qwen experiment was to show that these methods generalize across model families. Since receiving this feedback, we have gone one step further and replicated experiments on each of the datasets with Qwen 2.5 (Figure 6), finding nearly identical results. Additionally, we have also included an experiment where we use Llama 70B as the target model and compare Llama 8B, Qwen 7B, and Mistral 7B as proxy models (Figures 7-10). We find that using a proxy model from the same model family as the target model performs best, providing evidence that a similar training distribution between the target and proxy model is important for accurate proxy modeling.

---

### Official Review · Reviewer_Jn65 · 2024-11-04

**Soundness:** 2
**Presentation:** 3
**Contribution:** 3
**Rating:** 6
**Confidence:** 4

**Summary:**

The paper proposes a novel techniques called AttriBoT, which is an efficient approach for computing context attribution in LLMs by approximating the computationally expensive leave-one-out (LOO) error. It utilizes KV cache, hierarchical attribution, and proxy models to reduce the cost of calculating LOO attributions by over 300×, achieving real-time interpretability in context-augmented LLMs. The method provides a practical, feasible solutions for efficient real-world context attribution methods.

**Strengths:**

1. Efficient Context Attribution: The paper introduces an interesting and efficient approach to approximate the LOO error for context attribution from the perspective of cache reusage, hierarchical attribution, and smaller proxy model.

2. The  proposed different approaches can be composed together to achieve better efficiency and performance.

3. The framework shows strong performance across large language models by achieving over a 300× speedup in context attribution computations.

4. The paper also shows the implementation of efficient and easy-to-use AttriBoT.

**Weaknesses:**

1. The paper lacks an in-depth comparison with the latest methods, e.g., for SIG[1], a sequential-gradients-based approach to compute word importance, it used Log-Odds, Comprehensiveness, and Sufficiency evaluation metrics; for LOGRA[2], it evaluated the effectiveness in terms of accuracy and efficiency. However, these did not mention the LOO error. Could you please provide a more in-depth explanation of the necessity and importance of using the LOO error?

[1]. Enguehard, Joseph. "Sequential Integrated Gradients: a simple but effective method for explaining language models." arXiv preprint arXiv:2305.15853 (2023).

[2]. Choe, Sang Keun, et al. "What is Your Data Worth to GPT? LLM-Scale Data Valuation with Influence Functions." arXiv preprint arXiv:2405.13954 (2024).

2. The innovation is incremental, it has already well-established in large model compression and optimization methods for KV reuse, Mixture of Experts approach, and pruning strategies. Using cached activation avoiding redundant operation, hierarchical attribution reducing computation, and smaller proxy model emulating large model to approximate LOO seems to be a combination of the previous compression methods.

3. In the Approximation Error part. The paper focus on only a few highly contributive sources, which may overlook the cumulative effect of less influential spans. This approach might limit the method's generalizability, especially in cases where there isn't a clear distinction between highly and moderately contributive sources. It would be better to evaluate your method recovering few highly cotributive sources and full sources on datasets like HotpotQA to verify the rationality.

4. The experimental section needs better reorganization. Key experimental results like the  efficiency vs. accuracy trade-off for each AttriBoT acceleration method should be presented clearly within the main text.

**Questions:**

Please refer to Weakness 3 in the section above.

---

> ### Author Response · Authors · 2024-11-22
> **Author Response**
>
> Thank you for taking the time to read our paper and for providing this thoughtful response. Below, we provide some further context and discuss the weaknesses identified in your review.
>
> # Importance of the LOO Error and its approximation
> The LOO error measures the change in a model’s prediction when a single piece of data (e.g., one training example, one context source, or other notions of a single datum) is removed from the model. This provides a simple and interpretable score measuring how important that piece of data is to a model’s prediction or behavior. In fact, approximating the LOO error is the objective of many methods that aim to attribute back to training data (e.g., including LOGRA, but also datapoint attribution methods like influence functions and TRAK) or to context sources (e.g., ContextCite).
>
> In training data attribution works, such as the LOGRA paper highlighted in the review, computing the exact LOO error for a sample requires re-training the model with that sample excluded from the training set, which makes computing the exact LOO error for every training example infeasible. In fact, the basis of many training data attribution methods, like LOGRA, is to approximate the LOO error since it is not possible to compute directly. This is discussed in [Section 2: Scalability Bottlenecks in Influence Functions](https://arxiv.org/pdf/2405.13954) from the LOGRA paper.
>
> Since the exact LOO error is infeasible to compute in training data attribution, this also makes it prohibitively expensive to evaluate training data attribution methods in terms of how well they approximate the LOO error. As a result, most training data attribution methods resort to evaluating with proxy metrics like task accuracy as a function of the number of high-attribution training examples removed.
>
> For context attribution, the picture is a bit different. In this setting, it is expensive, *but feasible*, to compute exact LOO attributions since each attribution score requires a single forward pass rather than a full training run. Thus, in our work, we are actually able to measure how well our approximate LOO attribution methods align with the ground truth LOO attributions. To improve the clarity of the paper, we will be adding parts of this explanation to our related work.
>
> # Efficient LOO Attribution and Large Model Compression
> The goals of efficient LOO attribution and model compression do seem somewhat similar – in both settings we aim to emulate some behavior of large models efficiently. However, model compression focuses on retaining performance on downstream tasks while we primarily care about retaining the LOO attribution behavior of models. As such, there may be some similarity between the hierarchical, pruning, and proxy modeling approaches we propose and methods used for model compression, but we develop novel variants of these ideas and specifically test that the underlying assumptions of each of these methods hold true in the LOO attribution setting. Based on your feedback, we would be happy to add some references to the relevant literature on model compression, and if you have any suggestions for relevant papers, we would be happy to add them.
>
> # Recovering Attributions of More Than the Top Sources
>
> Most applications of context attribution (via LOO error or otherwise) primarily aim to identify highly contributive sources because the insight that many spans contributed a small amount is not particularly actionable. We note that AttriBoT will absolutely provide attribution scores for all sources in the context, but we focus on evaluating the highly contributive sources to ensure a reliable and realistic evaluation. If there is a different metric you would be interested in seeing on HotpotQA or some other dataset, we would be happy to compute and report it.
>
> # Organization of Experimental Results Section
>
> Due to space constraints, we had to defer some figures to the appendix while referring to them in the main text’s experimental results. We will try to make this section more clear, and if our submission is accepted, we plan to use the extra page to move these Appendix figures back into the main text as you've suggested.

---

> ### Comment · Reviewer_Jn65 · 2024-11-26
>
> Thanks for the authors' detailed responses. My main concerns have been addressed. Therefore, I am happy to maintain my positive rating.

---

### Official Review · Reviewer_abNN · 2024-11-09

**Soundness:** 3
**Presentation:** 3
**Contribution:** 3
**Rating:** 6
**Confidence:** 4

**Summary:**

The paper presents AttriBoT, a set of techniques including key-value caching, hierarchical attribution, and proxy modeling/pruning, to efficiently approximate leave-one-out context attribution for large language models, achieving significant speedup and better faithfulness compared to prior methods, with implications for LLM interpretability and various applications.

**Strengths:**

* The paper presents a novel approach, AttriBoT, for efficiently approximating the Leave-One-Out (LOO) error in context attribution for large language models (LLMs). The idea of using cached activations to avoid redundant operations is highly original. By caching the attention key and value tensors at each layer, the method significantly reduces the computational cost, which is a new way of addressing the inefficiency issue in computing LOO attributions. This has not been explored in such a comprehensive manner in previous works.
* In the context of the increasing use of LLMs, understanding how the model generates its output based on the input context is crucial. The work on context attribution, especially the proposed AttriBoT method, is highly significant as it enables more efficient analysis of the influence of each context span on the LLM's generations. This has practical implications in various applications such as improving the reliability and safety of LLMs by detecting malicious prompts and model hallucinations.
* The hierarchical attribution technique is a novel addition. The assumption that the sum of the LOO attributions for $k$ contiguous text spans can be approximated by a single Leave-$k$-Out attribution score is an innovative concept. This allows for a reduction in the number of forward passes required for attribution computation, especially in hierarchical contexts like paragraphs and sentences.

**Weaknesses:**

__theoretical analysis__
* For the similarity assumption between the proxy model and the target model, the paper mainly proves it by experimentally measuring the correlation, but does not deeply explore the stability and limitations of this similarity assumption under different model architectures and training data distributions. In some cases, the small proxy model may not accurately capture the complex behavior of the large target model, leading to deviations in the attribution results.

__generalization__
* Investigate the adaptability of the method to different model architectures. Consider testing on models with different architectural designs (such as models with different attention mechanisms, non - Transformer-based models) to see if the proposed techniques can be generalized or need to be adjusted. This will provide more comprehensive guidance for the application of the method in the broader field of LLMs.

__evaluation and baselines__
* Explore ways to integrate context attribution results into the human-machine interaction process. For example, design a feedback mechanism that provides users with context attribution information when they receive a response from the LLM, helping them to better understand the basis of the model's answer and guiding them to ask more effective questions. This can improve the overall user experience and the effectiveness of using LLMs.
* The choice of baselines is relatively limited. Although several common methods are included, there may be other emerging or less well-known methods that could provide a more comprehensive comparison. Additionally, the baselines may not cover all possible alternative approaches, leaving room for a more thorough evaluation of the novelty and superiority of the AttriBoT method.

**Questions:**

* When using proxy models for approximation, what is the impact of differences in the training data distribution between the proxy model and the target model on the accuracy of the context attribution? How can this potential issue be mitigated, especially when dealing with models trained on diverse or domain-specific datasets?
* In the hierarchical attribution method, how do you ensure the stability and accuracy of the approximation when the context structure becomes extremely complex, such as in documents with highly nested or irregular hierarchical structures? Are there any theoretical guarantees or additional techniques that could be employed to handle such cases?

---

> ### Author Response · Authors · 2024-11-22
> **Author Response**
>
> Thank you for taking the time to review our work. We agree that context attribution is a topic of crucial importance for understanding how LLMs “think” as well as allowing us to easily verify and validate their context-dependent claims. Below, we would like to discuss the three weaknesses you identified in your review:
>
>
> # Analysis of Proxy Model Fidelity
>
> We agree that the AttriBoT methods that rely on proxy modeling degrade as the similarity between the proxy model and the target model decreases. This is seen in our results where larger proxy models achieve higher context attribution fidelity than smaller proxy models (Figure 3). Spurred by your suggestions, we have expanded on these results by investigating how differences in training distribution between the target and proxy model affect context attribution. In particular, we evaluate the use of Llama 8B, Qwen 7B, and Mistral 7B as a proxy for Llama 70B and find the LOO attribution approximation error is higher with proxy models from a different model family than the target model (mAP of 0.81 for Llama 8B vs. 0.75 Mistral 7B and 0.71 for Qwen 7B). This gives evidence that matching the training distribution between the target and proxy model is important, and we have added these results to our paper (Figures 7-10, appendix). However, given that most LLM model families include models of various sizes (e.g. the Llama 3 collection now contains 5 models from 1B to 405B parameters), we think practitioners will likely be able to choose a proxy model from the same family.
>
> # Generalization to other model classes
>
> We focus on decoder-only transformers in this work for two reasons. First, nearly all LLMs are decoder-only transformers, and second, many decoder-only transformers have been trained at a variety of model sizes allowing for methods that leverage multiple models from the same family. To maximize the impact of our work, we chose to focus on this model class. However, if the reviewer has a specific suggestion of other language model architectures that are state-of-the-art and come in multiple sizes, we would certainly be interested in evaluating them too.
>
> # Incorporating Context Attribution into the Human-Machine Interaction Process
>
> This would be a useful and interesting application of our research on efficient LOO context attribution. Our specific goal in this paper was to lay the algorithmic foundation for an application like the one you've proposed. However, we will be open sourcing a library (see supplementary files for a recent version) with an easy-to-use API for performing LOO attribution with the AttriBoT set of methods. We hope that this will empower practitioners to enhance the usability of LLMs deployed in their products.
>
> # Baselines
> We believe we have covered all relevant baselines (5 including standard LOO) for context attribution. If the reviewer has any specific suggestions on additional baselines for context attribution, we would be happy to test them against our existing baselines and AttriBoT.

---

> > ### Comment · Reviewer_abNN · 2024-11-26
> >
> > Thanks for responding to my previous concerns and pointing out my misunderstanding in my original comments. The main issue of the proxy model has been well explained. I will keep my original rating.

---

### Author Response · Authors · 2024-11-22
**Updates to our Submission**

Thank you to all seven reviewers for engaging with our work and providing insightful feedback. Here, we would like to update you all on the changes we’ve made to our submission in response to your comments:

- We updated Figure 2 to indicate which combination of methods achieved different points on the AttriBoT Pareto front
- We’ve added a “Practitioner’s Guide to Using AttriBoT” in the Appendix to advise users on how best to apply AttriBoT to their problem and select appropriate algorithms and hyperparameters
- We expanded our set of results by evaluating our approximate LOO attribution methods using the Qwen 2.5 model family on all three datasets we consider in the paper (see Figure 6)
- We added experiments exploring the effectiveness of proxy modeling when the target model and proxy model come from different model families – e.g., Llama 70B approximated by Mistral 7B (see Figures 7-10)
- We have made some changes to our organization and notation to improve the paper’s clarity where indicated by reviewers

---

### Meta-Review · Area_Chair_vwTS · 2024-12-17

**Metareview:**

This paper simultaneously achieves a big speedup and also faithfull attributions compared to existing work.  The reviewers unanimously vote for acceptance, and I agree.  The reviewers did raise some concerns, but none of these were disqualifying, and the authors engaged extensively with the reviewers during the discussion period, including additional experiments.  For example, several reviewers raised novelty concerns, but methods need not be entirely novel as long as they obtain stronger results and are often composed of existing techniques.  Reviewers also pointed out that the method focuses on decoder-only transformers, which I think is entirely acceptable since virtually all LLMs in use are decoder-only transformers.  Overall, the reviewers and I have a positive view of this paper.

**Additional Comments On Reviewer Discussion:**

The reviews were already positive before rebuttals, but nonetheless, the authors did a great job during rebuttals addressing feedback.

---

### Decision · Program_Chairs · 2025-01-22

Accept (Poster)